# Characterization of a Major Quantitative Trait Locus for the Whiteness of Rice Grain Using Chromosome Segment Substitution Lines

**DOI:** 10.3390/plants13243588

**Published:** 2024-12-23

**Authors:** Lulu Chen, Yujia Leng, Caiyun Zhang, Xixu Li, Zhihui Ye, Yan Lu, Lichun Huang, Qing Liu, Jiping Gao, Changquan Zhang, Qiaoquan Liu

**Affiliations:** 1Jiangsu Key Laboratory of Crop Genomics and Molecular Breeding/Jiangsu Co-Innovation Center for Modern Production Technology of Grain Crops, Agricultural College, Yangzhou University, Yangzhou 225009, China; chenlulu9902@163.com (L.C.); yujialeng@yzu.edu.cn (Y.L.); 18361310213@163.com (C.Z.); lxx012179@163.com (X.L.); 19352979818@163.com (Z.Y.); luyan@yzu.ed.cn (Y.L.); 2Key Laboratory of Plant Functional Genomics of the Ministry of Education/Zhongshan Biological Breeding Laboratory, Yangzhou University, Yangzhou 225009, China; lchuang@yzu.edu.cn (L.H.); qliu2102002@gmail.com (Q.L.); 007446@yzu.edu.cn (J.G.); 3Provincial Key Laboratory of Agrobiology, Jiangsu Academy of Agricultural Sciences, Nanjing 210014, China

**Keywords:** rice, whiteness, quantitative trait loci, chromosomal segment substitution lines

## Abstract

The whiteness of rice grains (WRG) is a key indicator of appearance quality, directly impacting its commercial value. The trait is quantitative, influenced by multiple factors, and no specific genes have been cloned to date. In this study, we first examined the correlation between the whiteness of polished rice, cooked rice, and rice flour, finding that the whiteness of rice flour significantly correlated with both polished and cooked rice. Thus, the whiteness of rice flour was chosen as the indicator of WRG in our QTL analysis. Using a set of chromosome segment substitution lines (CSSL) with *japonica* rice Koshihikari as the recipient and *indica* rice Nona Bokra as the donor, we analyzed QTLs for WRG across two growth environments and identified six WRG QTLs. Notably, *qWRG9* on chromosome 9 displayed stable genetic effects in both environments. Through chromosomal segment overlapping mapping, *qWRG9* was narrowed to a 1.2 Mb region. Additionally, a BC_4_F_2_ segregating population confirmed that low WRG was a dominant trait governed by the major QTL *qWRG9*, with a segregation ratio of low to high WRG approximating 3:1, consistent with Mendelian inheritance. Further grain quality analysis on the BC_4_F_2_ population revealed that rice grains carrying the *Indica*-type *qWRG9* allele not only exhibited lower WRG but also had significantly higher protein content. These findings support the fine mapping of the candidate gene and provide an important QTL for improving rice grain quality through genetic improvement.

## 1. Introduction

Rice serves as a staple food and essential nutritional component for over half of the global population, often forming a significant part of daily diets. It is predominantly consumed as polished white rice, making the quality of this form crucial in shaping dietary preferences. The appearance of rice grains is a primary factor that directly influences their market value, which is of considerable interest to producers, consumers, and breeders alike. Moreover, the appearance of rice grains impacts milling quality, as well as the cooking and eating quality of the rice. The rice grain appearance is a complex trait, encompassing factors such as grain shape, chalkiness, transparency, color, and the whiteness of both polished and cooked rice [1,2].

Grain shape, determined by length, width, and thickness, is a critical index for rice grain appearance quality, impacting both yield and quality [1]. Rice grain chalkiness, a significant appearance trait, affects consumer acceptance and is typically caused by the deposition of starch and storage proteins in the endosperm [3]. Additionally, grain transparency, which reflects how well light passes through the grain, has received significant attention in recent years. High-quality rice with low amylose content tends to be less transparent [4]. The grain color primarily pertains to brown rice, which contains pigment in the pericarp, whereas the endosperm of polished rice from natural rice germplasms remains unpigmented. The whiteness of rice grain (WRG) is another crucial trait influencing the external appearance of polished and cooked rice. However, research into WRG has progressed slowly due to its complex influencing factors. Studies have shown significant phenotypic variation in WRG among rice cultivars, with physicochemical properties such as amylose content, alkali spreading value, and protein content having a modest impact on WGR [2,5]. Additionally, storage conditions like temperature and duration, along with milling precision, are significant factors affecting WRG [6,7].

Unlike grain shape and chalkiness, for which numerous genes have been identified, research on the genetic basis of WRG is relatively sparse [8,9]. WRG is a quantitative trait controlled by multiple quantitative trait loci (QTLs), yet studies focusing on QTL mapping for WRG are limited [10]. Although consumer preferences often hinge on the overall appearance of white rice, including its whiteness, the genetic underpinnings remain underexplored. Utilizing doubled haploid rice lines, researchers have identified two QTLs associated with the whiteness of cooked rice (WCR), namely, *qWCR3* and *qWCR11.* Rice harboring both QTLs exhibits the highest WCR [11]. Additionally, studies on the whiteness of polished rice (WPR) have led to the identification of two QTLs, *qWPR1* and *qWPR4*, using recombinant inbred lines derived from a cross between two closely related *japonica* cultivars [6]. Moreover, Hori et al. identified eleven and six QTLs for WPR using chromosome segment substitution lines (CSSLs) in the “Koshihikari” and “Takanari” genetic backgrounds, respectively [10]. Despite these advances, no specific genes directly associated with WRG have been identified. Therefore, elucidating the genetic factors influencing WRG is crucial for enhancing the aesthetic and commercial qualities of rice, guiding future breeding programs aimed at improving this important trait.

CSSLs are a set of lines that are developed through multiple rounds of backcrossing and selfing, accompanied by molecular marker assisted selection [12,13]. CSSLs offer significant advantages for fine mapping and cloning QTLs due to their clean genetic backgrounds. For example, a set of CSSLs was developed using the japonica rice variety Koshihikari (KOS) as the recipient, and several QTLs associated with rice grain quality traits were identified, including grain chalkiness, gel consistency, amylose content, grain size, grain whiteness, and the eating quality of cooked rice grains [10,14]. Among these, the genetic effects of *qWH1*, which is responsible for high whiteness and high eating quality scores in the “Koshihikari” and “Takanari” genetic backgrounds, were confirmed. However, there is no further evidence regarding the stable inheritance of this QTL or its genetic effect in segregating populations.

To identify stable WRG-related QTLs and assess their genetic effects in segregating populations, this study utilized a CSSL population derived from crosses between the japonica rice cultivar KOS and the indica rice cultivar Nona Bokra (NONA) to analyze WRG-related QTLs under two distinct environments. Furthermore, the individual CSSL line N114, known to carry the major QTL influencing *qWRG9*, was backcrossed with the recipient parent. Through linkage analysis, the major QTL *qWRG9* was further validated. Additionally, based on the QTL analysis results, we explored the genetic relationships between WRG and protein content. These findings are anticipated to provide valuable insights for the fine mapping and cloning of *qWRG9*, as well as for marker-assisted selection aimed at improving rice grain quality.

## 2. Results

### 2.1. The Whiteness of Rice Grain in the CSSLs Population

When assessing the whiteness of rice grains, typical measures include the whiteness of both polished rice grains and cooked white rice. However, the accuracy of these assessments may be affected by several factors. For example, spaces between polished grains can skew results, and the cooking temperature can alter the apparent whiteness of cooked rice [15]. Therefore, using rice flour for whiteness measurements is preferred because its uniform texture offers a more consistent and accurate indicator of the rice grain’s whiteness. The initial comparison of whiteness was conducted between the parental rice varieties, KOS and NONA. Significant differences in grain appearance were observed, with KOS exhibiting a superior appearance in both polished and cooked rice forms compared with NONA (Figure 1A). Similarly, rice flour from KOS also showed a comparable improvement in appearance. When quantifying whiteness, KOS demonstrated higher values across all three parameters, including polished rice, cooked rice, and rice flour, than NONA (Figure 1B–D). Notably, the whiteness value of rice flour was significantly higher than that of the other two parameters, indicating its enhanced effectiveness in representing the true whiteness of rice. To further explore the relationships among these measures, 20 individuals from the CSSLs were randomly selected for detailed grain whiteness analysis. The results (Figure 1E,F) showed strong correlations between the whiteness of polished and cooked rice with the whiteness of rice flour. Thus, the whiteness of rice flour was chosen as the index to represent WRG for subsequent QTL mapping analysis.

### 2.2. Phenotypic Variation in the CSSL Population

To map the QTLs associated with WRG, the distribution of WRG was first analyzed in 143 lines of the CSSL population under two environmental conditions. As shown in Figure 2, WRG values for the CSSL population, measured in Hainan and Yangzhou in 2020, exhibited a continuous distribution, indicating that WRG was a typical quantitative trait. Due to the inability of NONA to grow in Yangzhou, seeds for parental comparison were not obtained from this location. However, data from the Hainan indicated that the whiteness of KOS rice was significantly higher than that of NONA. A broad range of variation in WRG was observed in the CSSL population under both environments, with greater variation detected in Hainan, suggesting that rice whiteness was influenced by environmental factors. Additionally, as the absolute values of skewness and kurtosis were less than 1, the CSSL population was regarded as suitable for QTL analysis (Table 1).

### 2.3. Identification QTLs for WRG

Based on population variation analysis, QTLs controlling WRG were identified in two environments. Six QTLs were detected in Hainan and Yangzhou, with *qWRG1.1* and *qWRG1.2* identified only in Hainan and *qWRG6* and *qWRG10* only in Yangzhou, suggesting that these QTLs were strongly influenced by environmental conditions (Figure 3). In contrast, *qWRG7* and *qWRG9* were detected in both environments, indicating stable inheritance with less environmental influence. LOD values for these QTLs ranged from 2.61 to 6.54, with corresponding phenotypic variations between 3.94% and 12.7% (Table 2).

A detailed analysis was conducted on the two stable QTLs. The *qWRG7* was mapped to a region of approximately 440 kb between molecular markers SSR7-16 and SSR1-145 (Table 2), with phenotypic variations of 3.94% and 12.7% and additive effect values of −2.07 and −1.67 in two environments, respectively. Rice varieties carrying this QTL exhibited reduced WRG. Further investigation within this region identified the major gene *Rc*, which controls red pericarp and is located near this interval [16]. The analysis of two CSSL lines with this QTL revealed the presence of a red pericarp, resulting in some red aleurone layer remaining in polished rice and thereby reducing rice flour whiteness. The second stable QTL, *qWRG9*, exhibited phenotypic variations of 5.12% and 6.54% and additive effect values of −1.26 and −1.24 across the two environments. CSSL lines carrying this QTL also displayed reduced WRG. Due to the larger mapping interval and limited molecular markers in this region, three CSSL lines carrying different *qWRG9* were selected for further analysis.

### 2.4. Verification of qWRG9

To validate the stability of the *qWRG9* locus, three CSSL lines, including SN114 (low WRG), SN115 (low WRG), and SN117 (high WRG), along with their parent varieties, were subjected to whole-genome resequencing. The analysis revealed that while each of the three lines contained at least two introduced chromosomal segments, all shared a common segment on chromosome 9. Through overlapping mapping of these segments, *qWRG9* was localized to a region of approximately 2.2 Mb at the end of chromosome 9 (Figure 4A). To assess the genetic effect of the *qWRG9* locus, the CSSL line SN114, which had the fewest introduced segments, was selected for further analysis. A BC_4_F_2_ backcross population of 143 individuals was generated from a cross between KOS and N114 and used for genetic analysis. A molecular marker, W9-1, located within the introgressed segments, was developed based on the resequencing data and was utilized to do the genetic linkage analysis. Meanwhile, the WRG values of these 143 individuals were also examined, revealing that low-whiteness rice grain exhibited a recessive pattern with a typical bimodal distribution (Figure 4B). The ratio of low-WRG to high-WRG individuals was approximately 109:34, consistent with the expected 3:1 segregation ratio for a single locus. Further genotyping of the 143 individuals was conducted, and a subset of 29 individuals with different *qWRG9* genotypes was analyzed for WRG comparison. The results showed that rice lines with the homozygous *qWRG9* allele from NONA exhibited low WRG, while those with heterozygous or homozygous *qWRG9* alleles from the KOS displayed relatively high WRG (Figure 4C). These findings suggested that the *qWRG9* locus was closely associated with the low-WRG trait and that this trait was controlled by the major dominant QTL *qWRG9* from indica rice NONA. The identification of the major QTL *qWRG9* provided critical insights for the future cloning of the candidate gene, making *qWRG9* a promising target for breeding high-quality rice cultivars.

### 2.5. Effect of qWRG9 on Rice Grain Quality Traits

To further evaluate the impact of the *qWRG9* locus on rice grain quality, ten individuals carrying the *qWRG9* locus and ten individuals without it were selected from the BC_4_F_2_ population using marker-assisted selection for quality analysis. First, the amylose content was measured, revealing no significant difference between the two groups (Figure 5A). This finding contrasted with previous studies that reported a negative correlation between amylose content and rice whiteness, possibly due to the relatively uniform genetic background of the plants used in this study. Similarly, analysis of the gelatinization temperature also indicated no significant difference between the two groups (Figure 5B). However, the protein content analysis revealed that rice carrying the *qWRG9* locus had a significantly higher protein content than control lines, suggesting that in addition to enhancing rice whiteness, *qWRG9* also influenced the protein content (Figure 5C). Finally, the total starch content analysis revealed no significant difference between the two groups, although rice carrying the *qWRG9* locus tended to have slightly lower total starch content (Figure 5D).

### 2.6. Effect of qWRG9 on Rice Grain Protein Components

It is well known that the rice protein content plays a dual role in determining rice grain nutritional quality and taste quality [17]. Recent studies have found a significant negative correlation between protein content and rice taste quality [18]. To further evaluate the impact of the *qWRG9* locus on rice storage proteins, ten individuals carrying different *qWRG9* genotypes were selected from the BC_4_F_2_ population for protein component analysis. First, protein components of milled rice flours from the parent varieties and representative individuals were compared using SDS-PAGE. Compared with the parent variety KOS, the indica variety NONA showed a slight increase in almost all protein bands (Figure 6A). Similarly, rice lines carrying the *qWRG9* N genotype accumulated a significantly higher level of protein bands than those carrying the N genotype. Then, the four component proteins (albumin, globulin, prolamin, and glutelin) were further measured, and the results showed that rice lines carrying the *qWRG9* N genotype from NONA had significantly increased levels of glutelin and albumin compared with the rice lines with the *qWRG9* K genotype from KOS (Figure 6B–E). The above results were largely consistent with the total protein determination results and further indicated that *qWRG9* not only had a significant impact on rice whiteness but also significantly affected the protein content of rice.

## 3. Discussion

The appearance quality of rice is a primary consideration for consumers, with whiteness being a key indicator [19]. Rice whiteness is a multifaceted trait encompassing the whiteness of milled rice, cooked rice, and rice flour. Traditionally, rice whiteness is assessed through sensory evaluation, which is heavily influenced by subjective factors [20]. Numerous studies have reported the use of instruments to measure rice whiteness. For instance, Lv et al. used a spectrophotometer to analyze parboiled rice whiteness, finding a strong correlation between the whiteness of parboiled rice flour and whole-grain whiteness [21]. Similarly, Goto et al. used a portable spectrophotometer to measure the sensory whiteness of cooked rice and rice cakes from various rice varieties, demonstrating its effectiveness for whiteness measurement [5]. In this study, a colorimeter was employed to assess the whiteness of milled rice, cooked rice, and rice flour from CSSLs, followed by a correlation analysis. The results indicated that rice flour whiteness strongly correlated with the whiteness of both milled and cooked rice. Since milled and cooked rice whiteness can be affected by the uniformity of rice grains and milling precision, rice flour, due to its uniform texture, provides a more stable measure of whiteness. Therefore, rice flour whiteness was selected as the representative parameter for rice whiteness traits, aiming to enhance the accuracy of QTL mapping.

The whiteness of rice is a quantitative trait governed by multiple factors [10]. Currently, there is limited genetic research on the regulation of rice whiteness. Although some studies have identified QTLs related to the whiteness of cooked and milled rice, no specific genes have been cloned thus far. This is likely due to the influence of numerous minor loci and the trait’s high sensitivity to environmental conditions. CSSLs represent a valuable genetic resource for investigating complex traits as they consist of introgression lines adapted to a common genetic background while retaining diverse genetic segments. Using a CSSL population, we identified six QTLs associated with rice whiteness across two different environmental conditions. Among these, only *qWRG7* and *qWRG9* showed stable inheritance in both environments. Analysis of the mapping interval and phenotypic data suggested that *qWRG7* may be linked to the *Rc* gene, which controls red pericarp [16]. Similarly, Hoai et al. used two CSSL populations with distinct genetic backgrounds to identify a stable whiteness QTL, *qWH7*, on chromosome 7 [10], further confirming the significance and consistency of this QTL. However, *qWRG9* has not been reported in previous studies, possibly due to differences in the donor rice varieties. In contrast, prior studies have identified QTLs for the whiteness of cooked rice on chromosomes 3 and 11, with a combined effect that enhances whiteness [11]. Additionally, QTLs for the whiteness of milled rice have been reportedly located on chromosomes 1 and 4 [6]. The QTLs identified in this study differed from those reported previously, highlighting the significant variation in whiteness-related QTLs among rice varieties and indicating that these traits were dependent on the genetic background of the specific variety.

Genetic research on grain whiteness in rice is lagging behind that of wheat, where multiple QTLs for flour whiteness have been identified, and genes related to color synthesis, such as Phytoene Synthase 1 (Psy1), play a significant role in influencing flour yellowness [22]. In rice, while some studies have examined gene functions, these have primarily focused on biotic and abiotic responses without reported impacts on rice whiteness, suggesting functional differences in genes between species. Notably, Hori et al. found that the QTL *qWH1* from *indica* rice not only enhanced grain whiteness but also improved eating quality [10]. In our study, we identified *qWRG9* as a stably inherited QTL across different environments. More importantly, genetic linkage analysis revealed that *qWRG9* can be transmitted in offspring as a single Mendelian factor, exhibiting incomplete dominance. This suggests that *qWRG9* can be finely mapped and cloned using map-based cloning methods. Additionally, molecular markers closely linked to *qWRG9* can be effectively used in marker-assisted breeding for rice quality improvement.

The appearance quality of rice encompasses multiple factors, including grain shape, chalkiness, whiteness, and transparency. However, research on the mechanisms underlying whiteness formation in rice remains limited [1]. Earlier studies primarily focused on factors influencing rice whiteness, revealing that it is closely related to storage temperature, milling degree and duration, cooking time, and moisture content [23,24]. Additionally, endosperm composition, particularly amylose and protein content, can potentially impact rice whiteness. The donor parent, NONA, carries the high amylose gene *Wx^a^*, resulting in some CSSL lines with high amylose content. However, this locus was not identified in the whiteness QTL mapping, indicating that high amylose content did not affect rice whiteness in the KOS background. Notably, rice lines carrying the *qWRG9* locus exhibit increased protein content, consistent with findings that rice whiteness is negatively correlated with protein content [5]. This suggests that *qWRG9* from KOS may enhance rice whiteness and eating quality, similar to the effect of *qWH1* in *indica* rice [10]. The existing studies show that protein content is closely related to eating quality, with lower protein content generally associated with better eating quality [25,26]. As a well-characterized high-quality *japonica* rice, KOS’s superior appearance and eating quality may be attributed to the *qWRG9^KOS^* allele, which increases whiteness and reduces protein content. Thus, the superior allele type *qWRG9^KOS^* from KOS has significant breeding potential for improving rice whiteness and eating quality.

## 4. Materials and Methods

### 4.1. Plant Material

Koshihikari (KOS) a typical *japonica* rice variety with good grain quality, and the *indica* variety Nona Bokra (NONA), a typical *indica* variety from India, were used to develop a set of CSSLs. The CSSLs, derived with KOS as the recipient and NONA as the donor parent [27], consisted of 143 lines specifically selected for analyzing WRG. All the rice varieties and lines were cultivated under uniform conditions at the experimental farm of Yangzhou University in Yangzhou, Jiangsu Province, and Lingshui in Hainan Province, China, during the 2022 growing season. The CSSL population was planted in randomized block design with three replicates. Each replicate comprised 40 seedlings per line. The management of these field experiments adhered to standard agricultural practices to ensure consistent growth conditions and reliable data collection.

### 4.2. Methods

#### 4.2.1. Construction of the Mapping Population

For the genetic analysis of *qWRG9*, the CSSL line N114 was backcrossed with the recipient parent KOS, resulting in a BC_4_F_2_ population comprising 143 individuals. To assess the impact of *qWRG9* on rice grain quality traits, 20 individuals harboring different *qWRG9* genotypes were selected based on molecular marker selection from the BC_4_F_2_ generation for further analysis of grain quality.

#### 4.2.2. Identification of QTLs

For the identification of QTLs associated with WRG, 126 previously developed polymorphic markers were employed [27]. The analysis of these QTLs was conducted using IciMapping 4.2 software, which utilized an inclusive composite interval mapping (ICIM) approach. A LOD (logarithm of the odds) score threshold of 2.5 was set to determine significant QTLs [28]. Additionally, the additive effects, genetic parameters, and the percentage of phenotypic variation explained by each QTL were calculated. The nomenclature for the QTLs adhered to the method described by McCouch et al. [29]. The genetic map was constructed using MapChart version 2.32 software [30].

#### 4.2.3. DNA Extraction and Genotyping

Genomic DNA were extracted from rice leaves using the cetyltrimethylammonium bromide (CTAB) method [31]. For genotyping, genomic DNA samples from CSSL lines SN114, SN115, and SN117, along with KOS and NONA, were sent to MolBreeding Corporation (Shijiazhuang, Hebei Province, China) for next-generation sequencing on the MGI-2000/MGI-T7 sequencing platform (MGI Tech, Shenzhen, China). The sequencing was conducted at an average depth of 50×, and single nucleotide polymorphisms (SNPs) were mapped along the chromosomes according to their physical locations. For genetic linkage analysis, a molecular marker, W9-1 (F: 5′-TTCTCTTTATCAATTGCTGCGGC-3′; R: 5′-CCATCGTTGCTGCTGTTGGTT-3′), was developed based on the resequencing data. This marker was located within the introgressed segments and was utilized to facilitate the identification of genetic linkages. All molecular marker analyses were performed using the conventional PCR method.

#### 4.2.4. Evaluation of Whiteness of Rice

To assess the whiteness of rice grain, paddy rice was dehusked and milled to produce polished white rice with a grain polisher (Kett, Tokyo, Japan). To avoid interference from residual aleurone on the determination of rice whiteness, the milling time for all samples was extended from the normal sample’s 2 min to 3 min. Additionally, a portion of these polished grains was further processed using a FOSS™ 1093 Cyclotec Sample Mill (Foss Tecator, Höganäs, Sweden) to produce rice flour. The whiteness of the polished white rice, cooked white rice, and white rice flour was then measured using a Caipu DS-700D colorimeter (Hangzhou Caipu, Hangzhou, China). Each sample type, including the polished rice, cooked white rice, and white rice flour, was placed in a 30 mL white plastic box for measurement. The color parameters were assessed based on the Hunter system, specifically, the L*, a*, and b* values. Whiteness was calculated using the formula W = 100 − [(100 − L*)^2^ + a*^2^ + b*^2^]^1/2^, as described in a previous study [19].

#### 4.2.5. Analysis of Rice Grain Physicochemical Properties

The amylose content was measured using a modified iodine colorimetry method as described by Lu et al. [32]. Briefly, 50.0 mg of rice flour was gelatinized in a NaOH solution in a 2 mL centrifuge tub. Following gelatinization, a sodium acetate solution (pH 4.3) and an iodine solution (0.02%) were added. The absorbance of this mixture was then measured at 620 nm. The amylose content was calculated using a standard curve prepared from amylose standards with varying concentrations. The crude protein content was measured using an A Kjeltec 2300 nitrogen determination instrument (Foss Tecator) according to the standard procedures, utilizing a conversion factor of 5.4 to calculate the protein content. The gelatinization temperature of the rice starch was assessed using a DSC 200 F3 differential scanning calorimetry (DSC) instrument (Netzsch Instruments NA LLC, Burlington, MA, USA) as previously described [4]. Total proteins were extracted from the milled rice flour using the method of Yamagata et al. [33], and SDS-PAGE was performed using standard procedures. Gels were stained with Coomassie Brilliant Blue R250 to examine the protein bands. For the protein component analysis, four kinds of storage protein were extracted from the rice flours and quantified by means of the Bradford assays as described previously [34,35].

#### 4.2.6. Data Analysis

Phenotypic data from the CSSL population were organized and processed using Microsoft Excel 2021. Statistical analysis, including Student’s *t*-test and one-way analysis of variance (ANOVA), was conducted using SPSS 16.0 statistical software to evaluate differences and relationships within the data. Pearson’s bivariate correlations were also calculated using the same software to examine the associations between variables. The results were reported as means ± standard deviation (SD). Statistical significance was established at a probability value of *p* < 0.05. Each data point represented the average of three biological replicates to ensure reliability and accuracy in the reported findings.

## 5. Conclusions

This study investigated the whiteness of rice grains (WRG), a critical factor in the appearance quality and commercial value of rice. Despite being a quantitative trait influenced by multiple factors, no specific genes associated with WRG have been cloned. We found a significant correlation between the whiteness of rice flour and both polished and cooked rice, leading them to use rice flour whiteness as an indicator for WRG in their QTL analysis. Utilizing chromosome segment substitution lines (CSSLs) with japonica rice KOS and indica rice NONA, we identified six WRG QTLs across two growth environments, with *qWRG7* and *qWRG9* showing stable genetic effects. *qWRG9* was mapped to a 1.2 Mb region and proved that the low WRG was a dominant trait controlled by *qWRG9*, following a 3:1 segregation ratio. Importantly, we found that rice grains with the japonica-type *qWRG9* had better WRG and lower protein content, which may be the main factors contributing to KOS’s superior appearance and eating quality. The research also provides a foundation for further studies on the genetic basis of rice grain appearance quality, which could lead to more targeted and efficient breeding strategies.

## Figures and Tables

**Figure 1 plants-13-03588-f001:**
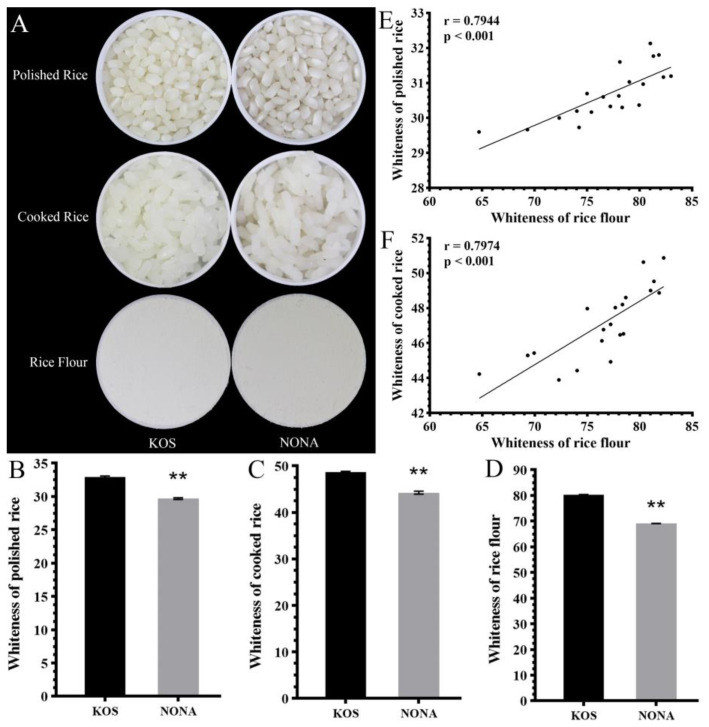
Grain whiteness of parental rice varieties and correlation analysis of whiteness parameters. (**A**) Appearance of polished rice, cooked rice, and rice flour from the parent varieties Koshihikari (KOS) and Nona Bokra (NONA). (**B**–**D**) Whiteness values of polished rice (**B**), cooked rice (**C**), and rice flour (**D**) for KOS and NONA rice varieties, with significant differences indicated (** *p* < 0.01). (**E**,**F**) Correlation analysis showing the relationship between the whiteness of the rice flour and the whiteness of polished rice (**E**) and cooked rice (**F**) with correlation coefficients and significance values displayed.

**Figure 2 plants-13-03588-f002:**
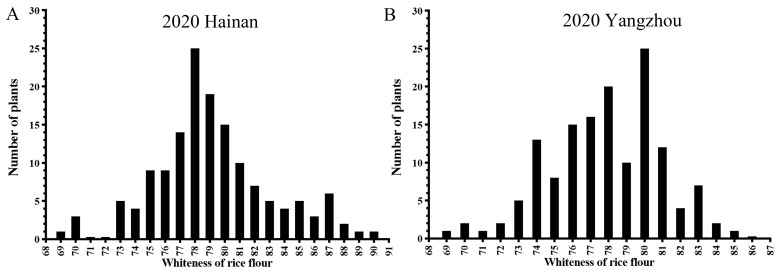
Frequency distributions of the whiteness of rice grains (WRG) in chromosomal segment substitution line (CSSL) populations under two environmental conditions. (**A**) Distribution of WRG in the CSSL population in Hainan, 2020. (**B**) Distribution of WRG in the CSSL population in Yangzhou, 2020.

**Figure 3 plants-13-03588-f003:**
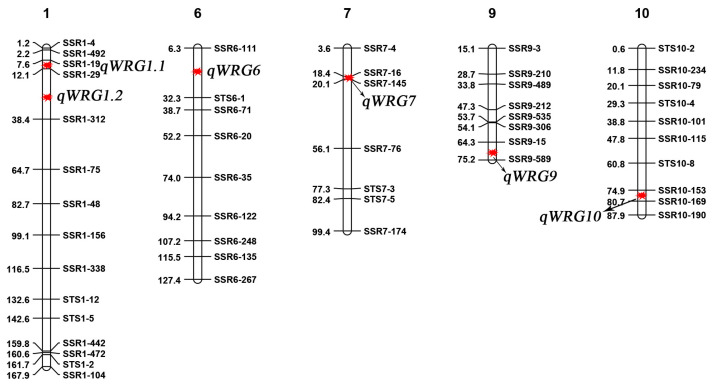
Locations of QTLs for the whiteness of rice grain (WRG) in the chromosomal segment substitution line (CSSL) populations across two environments. DNA markers with genetic distances (in centimorgans) are displayed along each chromosome. QTLs for WRG, including *qWRG1.1*, *qWRG1.2*, *qWRG6*, *qWRG7*, *qWRG9*, and *qWRG10*, are indicated by red markers.

**Figure 4 plants-13-03588-f004:**
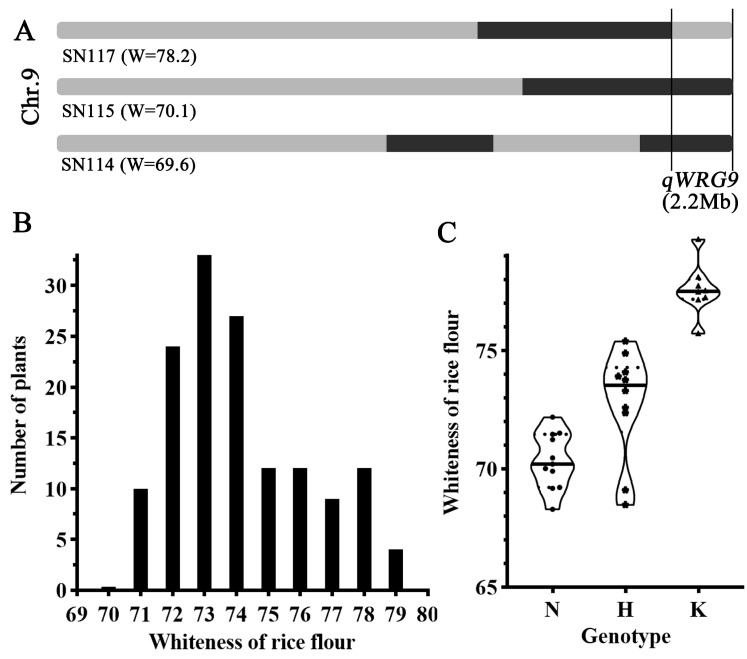
Contig mapping of *qWRG9* and frequency distributions of the whiteness of rice grain (WRG) in the BC_4_F_2_ population. (**A**) Localization of *qWRG9* on chromosome 9, based on introduced chromosomal segments in chromosome segment substitution lines (CSSLs) SN114, SN115, and SN117. The whiteness values (W) of each line are indicated in parentheses. (**B**) Frequency distribution of WRG in the BC_4_F_2_ population. (**C**) Comparison of WRG among genotypes: N indicates rice lines with the homozygous *qWRG9* allele from NONA. H indicates rice lines with the heterozygous *qWRG9* allele, and K indicates rice lines with the homozygous *qWRG9* allele from KOS.

**Figure 5 plants-13-03588-f005:**
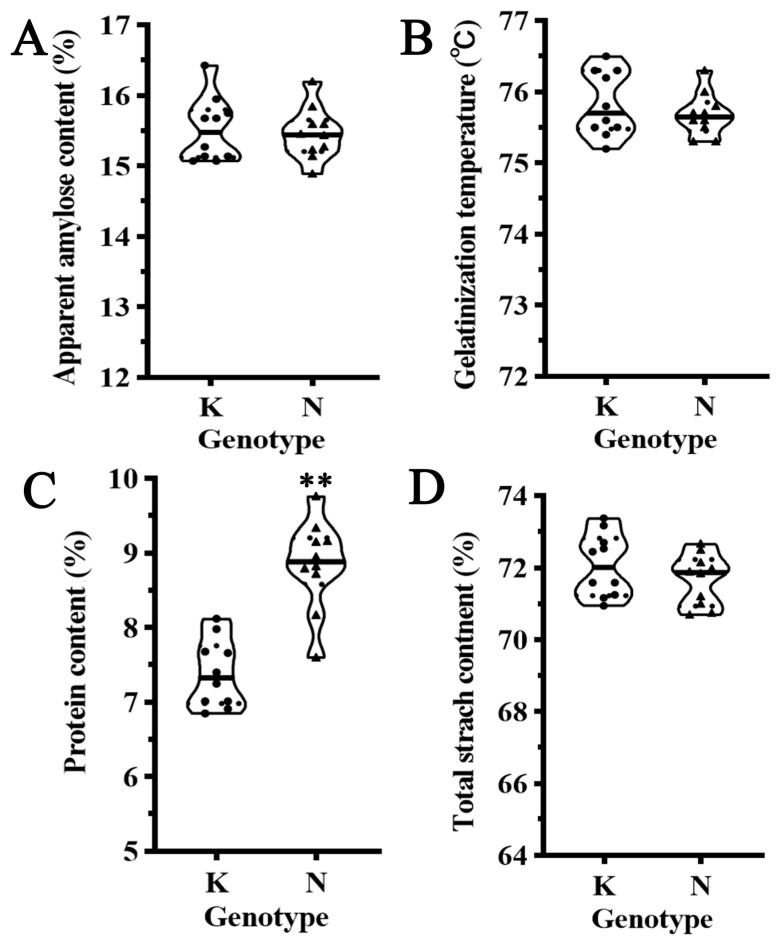
Rice grain quality traits in individuals with different *qWRG9* genotypes. (**A**) Apparent amylose content (%) in rice lines carrying the homozygous *qWRG9* allele from Koshihikari (K) and Nona Bokra (N). (**B**) Gelatinization temperature (°C) of rice starch in K and N genotypes. (**C**) Protein content (%) in K and N genotypes, showing a significant increase in individuals carrying the *qWRG9* allele from NONA (** *p* < 0.01). (**D**) Total starch content (%) in K and N genotypes.

**Figure 6 plants-13-03588-f006:**
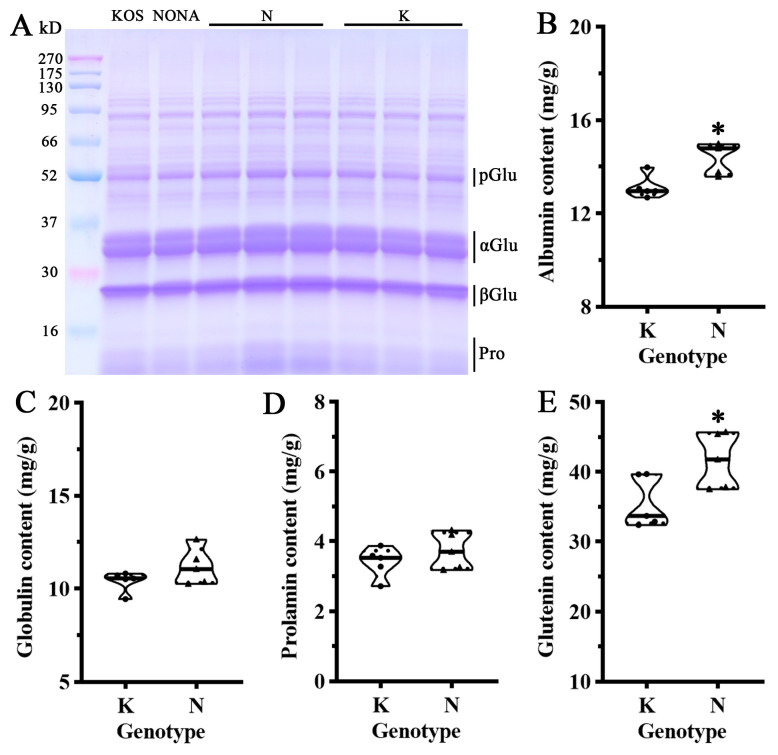
The comparison of protein components of milled rice with different *qWRG9* genotypes. (**A**) SDS-PAGE profiles of total storage from three representative rice flours in K and N genotypes, respectively. (**B**–**E**) The albumin, globulin, prolamin, and glutelin contents of five representative milled rice flours in K and N genotypes, respectively (* *p* < 0.05).

**Table 1 plants-13-03588-t001:** Description of the whiteness of rice grains (WRG) of two parental varieties and their chromosomal segment substitution line (CSSL) populations under two environmental conditions.

Year and Location	Parents			CSSLs Population			
	Koshihikira	Nona Bokra	*p* Value	Range	Mean ± SD	Skewness	Kurtosis
2020 Hainan	79.21	69.14	<0.0001	69.14~89.86	79.13 ± 4.07	0.3	0.36
2020 Yangzhou	77.73	─	─	69.41~85.31	77.78 ± 3.07	−0.25	−0.13

**Table 2 plants-13-03588-t002:** Description of the whiteness of rice grain (WRG) of parental varieties and their chromosomal segment substitution line (CSSL) populations across two environments.

Location	QTLs	Chr.	Markers	Position (Mb)	LOD	Additive Effect	PVE (%)
Hainan	*qWRG1.1*	1	SSR1-19–SSR1-29	1.90–3.03	3.38	1.43	7.16
	*qWRG1.2*	1	SSR1-29–SSR1-312	3.03–9.60	2.93	2.96	4.52
	*qWRG7*	7	SSR7-16–SSR1-145	4.59–5.03	4.64	−2.07	3.94
	*qWRG9*	9	SSR9-15–SSR9-589	16.08–18.81	5.12	−1.26	8.32
Yangzhou	*qWRG6*	6	SSR6-111–STS6-1	6.25–32.33	2.61	1.077	5.37
	*qWRG7*	7	SSR7-16–SSR1-145	4.59–5.03	6.41	−1.67	12.7
	*qWRG9*	9	SSR9-15–SSR9-589	16.08–18.81	6.54	−1.24	7.32
	*qWRG10*	10	SSR10-153–SSR10-169	18.7–20.1	3.74	−0.71	6.94

LOD: logarithm of odds; PVE: phenotypic variance explained.

## Data Availability

The data presented in this study are available on request from the corresponding author.

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
