# Peer review of "Characterization of a Major Quantitative Trait Locus for the Whiteness of Rice Grain Using Chromosome Segment Substitution Lines"

_plants, 2024, doi:10.3390/plants13243588_

Round 1
Reviewer 1 Report
Comments and Suggestions for Authors
Dear Authors,
The general idea of the manuscript “Characterization of a major QTL for the whiteness of rice grain (WRG) using chromosome segment substitution lines” is interesting. The manuscript made a very positive impression. The range of methods, topics and level of relevance of the results obtained correspond to the journal.
However, I have an important remark. There is clearly a problem with the References in the work. Due to this confusion, it is difficult to assess the level of novelty.
Best regards
Author Response
Comments: The general idea of the manuscript “Characterization of a major QTL for the whiteness of rice grain (WRG) using chromosome segment substitution lines” is interesting. The manuscript made a very positive impression. The range of methods, topics and level of relevance of the results obtained correspond to the journal.
Response: Thanks for the positive comments.
However, I have an important remark. There is clearly a problem with the References in the work. Due to this confusion, it is difficult to assess the level of novelty.
Response: Thanks for the good suggestion, and we sincerely apologize for the incorrect citation of references. We have carefully checked the references throughout the manuscript and made the necessary corrections. Additionally, based on the suggestions of other reviewers, we have added some new references.
Reviewer 2 Report
Comments and Suggestions for Authors
I checked your manuscript and described comments below.
In some countries, the whiteness of polished rice grains is an important issue that affects commercial value.
In this paper, a set of chromosome segment substitution lines (CSSLs) was created using Japonica rice as the recipient and Indica rice as the donor, and excellent QTL analysis was performed.
I think it would be better to consider the following points.
1. I think Koshihikira, which is Japonica rice, is a mistake for Koshihikari.
2. The whiteness of rice grains depends on how much it is polished. There is no description of the standard for this.
3. I think the QTL analysis in this paper is very good. I wish they had also analyzed correlations with nutritional value, etc.
I don't think this paper has major problems and grammatical problems.
Author Response
Commetns: I checked your manuscript and described comments below. In some countries, the whiteness of polished rice grains is an important issue that affects commercial value.In this paper, a set of chromosome segment substitution lines (CSSLs) was created using Japonica rice as the recipient and Indica rice as the donor, and excellent QTL analysis was performed. I think it would be better to consider the following points.
Response: Thanks for the positive comments.
Commetns: 1. I think Koshihikira, which is Japonica rice, is a mistake for Koshihikari.
Response: Thank you very much for the reviewer's reminder, and in deed, the Japonica rice used in the present study is “Koshihikari” but not “Koshihikira”, and we have correct it.
Commetns: 2. The whiteness of rice grains depends on how much it is polished. There is no description of the standard for this.
Response: Thanks for the good suggestion, and in deed, the whiteness of rice grains depends on how much it is polished. Usually, for brown rice, milling for 2 minutes can yield clean white rice, but in this experiment, we extended the milling time to 3 minutes to minimize the interference of incomplete milling on the determination of rice whiteness. We have added the description of the standard for this at page 11, lines 379 to 382 as follows: “To avoid interference from residual aleurone on the determination of rice whiteness, the milling time for all samples was extended from the normal sample's 2 minutes to 3 minutes.”
Commetns: 3. I think the QTL analysis in this paper is very good. I wish they had also analyzed correlations with nutritional value, etc.
Response: Thanks for the good suggestion. As the reviewer pointed out, this study found that qWRG9 has a significant impact on both the rice grain whiteness and eating quality of rice. Furthermore, basic rice quality analysis was conducted on different genotypes of rice plants, and it was found that, in addition to rice whiteness, the protein content of rice was most affected by qWRG9. Since the protein content of rice is an important indicator for evaluating the nutritional quality of rice, we added some data on the analysis of rice protein components and found that the main protein components of rice lines carrying qWRG9 showed a significant decreased trend. The new results were added at page 8, lines 239 to 255 as follows: “It is well known that rice protein content plays a dual role in determining rice grain nutritional quality and taste quality [17]. Recent studies have found a significant negative correlation between protein content and rice taste quality [18]. To further evaluate the impact of the qWRG9 locus on rice storage proteins, ten individuals carrying different qWRG9 genotypes were selected from the BC4F2 population for protein component analysis. Firstly, protein components of milled rice flours from the parent varieties and representative individuals were compared using SDS-PAGE. Compared with the parent variety KOS, the indica variety NONA showed a slight increase in almost all protein bands (Figure 6A). Similarly, rice lines carrying the qWRG9 N genotype accumulated a significantly higher level of protein bands than those carrying the N genotype. Then, the four component proteins (albumin, globulin, prolamin, and glutelin) were further measured, and the results showed that rice lines carrying the qWRG9 N genotype from NONA had significantly increased levels of glutelin and albumin compared to the rice lines with the qWRG9 K genotype from KOS (Figure 6B-E). The above results are largely consistent with the total protein determination results and further indicate that qWRG9 not only has a significant impact on rice whiteness but also significantly affects the protein content of rice.”
Commetns: I don't think this paper has major problems and grammatical problems.
Response: Thanks for the positive comments.
Reviewer 3 Report
Comments and Suggestions for Authors
The paper of Chen et al., "Characterization of a major QTL for the whiteness of rice grain (WRG) using chromosome segment substitution lines" presents investigated the genetic factors influencing the whiteness of rice grains, an essential trait for rice quality. The authors used chromosome segment substitution lines (CSSL) derived from crosses between japonica and indica rice varieties to map quantitative trait loci (QTLs) associated with rice grain whiteness. Here are some of my comments and suggestions:
1. Line77-79 you can cite more updated references that have used CSSL in their study. I have one example below but please add additional references.
Mabreja, A.D.; Reyes, V.P.; Soe, T.K.; Shimakawa, K.; Makihara, D.; Nishiuchi, S.; Doi, K. Evaluation of Grain-Filling-Related Traits Using Taichung 65 x DV85 Chromosome Segment Substitution Lines (TD-CSSLs) of Rice. Plants 2024, 13, 289, doi:10.3390/plants13020289.2. Revise the last paragraph of the introduction Loine 76-89. Higlight the objective of the study. In this current version, it is difficult to fully catch your objectives.
3. Lines 92-97: Revise that statement. You may use this.
"When assessing the whiteness of rice grains, typical measures include the whiteness of both polished rice grains and cooked white rice. However, the accuracy of these assessments may be affected by several factors. For example, spaces between polished grains can skew results, and the cooking temperature can alter the apparent whiteness of cooked rice. Therefore, using rice flour for whiteness measurements is preferred because its uniform texture offers a more consistent and accurate indicator of the rice grain’s whiteness."
3. In the results section, the authors must name the SSR and STS markers that were used. This will help in future progress of this research by other researchers.
4. The discussion part seems to have a smaller font size. Please double-check and revise accordingly.
Author Response
Comments: The paper of Chen et al., "Characterization of a major QTL for the whiteness of rice grain (WRG) using chromosome segment substitution lines" presents investigated the genetic factors influencing the whiteness of rice grains, an essential trait for rice quality. The authors used chromosome segment substitution lines (CSSL) derived from crosses between japonica and indica rice varieties to map quantitative trait loci (QTLs) associated with rice grain whiteness. Here are some of my comments and suggestions:
- Line77-79 you can cite more updated references that have used CSSL in their study. I have one example below but please add additional references.
Mabreja, A.D.; Reyes, V.P.; Soe, T.K.; Shimakawa, K.; Makihara, D.; Nishiuchi, S.; Doi, K. Evaluation of grain-filling-related traits using Taichung 65 x DV85 chromosome segment substitution lines (TD-CSSLs) of rice. Plants 2024, 13, 289, doi:10.3390/plants13020289.
Response: Thanks for the positive comments, and we have added the new references to the manuscript.
Comments: 2. Revise the last paragraph of the introduction Loine 76-89. Higlight the objective of the study. In this current version, it is difficult to fully catch your objectives.
Response: Thanks for the suggestion, and we have rewrite this section, the new version was at page 2, lines 82 to 99, as follows: “CSSLs are a set of lines that are developed through multiple rounds of backcrossing and selfing, accompanied by molecular marker assisted selection [12,13]. CSSLs offer significant advantages for fine mapping and cloning QTLs due to their clean genetic backgrounds. For example, a set of CSSLs was developed using the japonica rice variety Koshihikari (KOS) as the recipient, and several QTLs associated with rice grain quality traits were identified, including grain chalkiness, gel consistency, amylose content, grain size, grain whiteness, and the eating quality of cooked rice grains [10, 14]. Among these, the genetic effects of qWH1, which is responsible for high whiteness and high eating quality scores in the ‘Koshihikari’ and ‘Takanari’ genetic backgrounds, were confirmed. However, there is no further evidence regarding the stable inheritance of this QTL or its genetic effect in segregating populations.
To identify stable WRG-related QTLs and assess their genetic effects in segregating populations, this study utilized a CSSL population derived from crosses between the japonica rice cultivar KOS and the indica rice cultivar Nona Bokra (NONA) to analyze WRG-related QTLs under two distinct environments. Furthermore, the individual CSSL line N114, known to carry the major QTL influencing qWRG9, was backcrossed with the recipient parent. Through linkage analysis, the major QTL qWRG9 was further validated. Additionally, based on the QTL analysis results, we explored the genetic relationships between WRG and protein content. These findings are anticipated to provide valuable insights for the fine mapping and cloning of qWRG9, as well as for marker-assisted selection aimed at improving rice grain quality.”
Comments: 3. Lines 92-97: Revise that statement. You may use this.
"When assessing the whiteness of rice grains, typical measures include the whiteness of both polished rice grains and cooked white rice. However, the accuracy of these assessments may be affected by several factors. For example, spaces between polished grains can skew results, and the cooking temperature can alter the apparent whiteness of cooked rice. Therefore, using rice flour for whiteness measurements is preferred because its uniform texture offers a more consistent and accurate indicator of the rice grain’s whiteness."
Response: Thank you very much for the help in the statement, and we have added it to the manuscript.
Comments: 4. In the results section, the authors must name the SSR and STS markers that were used. This will help in future progress of this research by other researchers.
Response: Thanks for the suggestion, and we have made it clear at page 6, lines 192 to 194, as follows: “A molecular marker, W9-1 located within the introgressed segments was developed based on the resequencing data, and was utilized to do the genetic linkages analysis.”
Comments: 4. The discussion part seems to have a smaller font size. Please double-check and revise accordingly.
Response: Thank you very much for the reviewer's reminder, and we read through the manuscript and revised it.
Round 2
Reviewer 3 Report
Comments and Suggestions for Authors
The authors are not giving much attention to revising this manuscript. [Error! Reference source not found.] is all over the place. Please double-check everything. As for my suggestions and comments, the authors have addressed it.
Comments on the Quality of English Language
NA
Author Response
Commetns: The authors are not giving much attention to revising this manuscript. [Error! Reference source not found.] is all over the place. Please double-check everything. As for my suggestions and comments, the authors have addressed it.
Reply: Thank you very much for pointing out the errors in the format of our paper. We apologize for not noticing the errors produced by the system when generating the PDF file. Upon careful review, we found that the errors in the references were caused by the improper use of citation software. We have carefully read through the entire text and corrected every reference citation.